# Thymus Surgery Prospectives and Perspectives in Myasthenia Gravis

**DOI:** 10.3390/jpm14030241

**Published:** 2024-02-23

**Authors:** Paul Salahoru, Cristina Grigorescu, Marius Valeriu Hinganu, Tiberiu Lunguleac, Alina Ioana Halip, Delia Hinganu

**Affiliations:** 1Department of Surgery I, Faculty of Medicine, “Grigore T. Popa” University of Medicine and Pharmacy, 700115 Iasi, Romania; paul.p.salahoru@umfiasi.ro (P.S.); cristina.grigorescu@umfiasi.ro (C.G.); tiberiu.lunguleac1@umfiasi.ro (T.L.); 2Department of Morpho-Functional Sciences I, Faculty of Medicine, “Grigore T. Popa” University of Medicine and Pharmacy, 700115 Iasi, Romania; hinganu.delia@umfiasi.ro; 3Department of Medical III, Faculty of Medicine, “Grigore T. Popa” University of Medicine and Pharmacy, 700115 Iasi, Romania; ioana-alina.grajdeanu@umfiasi.ro

**Keywords:** thymectomy, myasthenia gravis, thoracic surgery, acetylcholine, video-thoracoscopy (VATS)

## Abstract

The thymus is a lymphoid organ involved in the differentiation of T cells, and has a central role in the physiopathogenesis of Myasthenia Gravis (MG). This connection is proved by a series of changes in the level of neuromuscular junctions, which leads to a decrease in the amplitude of the action potential in the post-synaptic membrane. Because of this, the presence of anti-cholinergic receptor antibodies (AChR), characteristic of MG, is found, which causes the progressive regression of the effect of acetylcholine at the level of neuromuscular junctions, with the appearance of muscle weakness. The thymectomy is a surgical variant of drug therapy administered to patients with MG. In the case of patients with nonthymomatous MG, thymectomy has become a therapeutic standard, despite the fact that there is no solid scientific evidence to explain its positive effect. Videothoracoscopic surgery or robotic surgery led to a decrease in the length of hospital stay for these patients. This paper aims to synthesize the information presented in the literature in order to create a background for the perspectives of thymectomy.

## 1. Introduction

The thymus is a lymphoid organ involved in the differentiation of T cells. It is divided into two main cellular areas: the cortex and the medulla. It is composed of various stromal cells: thymic epithelial cells (TECs), fibroblasts, macrophages, dendritic cells (DCs), and myoid cells [1]. Medullary TEC plays an important role in tissue-specific antigen (TSA) presentation, essential for thymocyte selection and education in the discrimination of self-antigens and non-self-antigens [2]. Medullary TEC expresses the different subunits of AChR as TSA [3]. Myoid cells also express the different subunits of AChR and display a functional receptor [4]. In patients with AChR MG, the thymus is considered the pathologic effector organ [5,6].

Thymus changes are present in approximately 80% of patients with generalized MG. The most common change is thymic hyperplasia, seen in about half of patients with this form of MG [7].

Thymoma occurs in about 15% of cases. Thymic atrophy also occurs in 10–20% of cases. In patients with thymic atrophy, replacement adipose tissue or residual areas of thymic parenchyma are found [8].

The pathophysiology of MG is complex, and is represented by a decrease in the number of active nicotinic cholinergic receptors (Nm) at the level of the neuromuscular junction, which leads to a decrease in the amplitude of the action potential at the level of the post-synaptic membrane [9].

The presence of AChR antibodies, characteristic of MG, causes the progressive regression of the effect of acetylcholine at the level of the neuromuscular junction [10].

As a result, there is an insufficiency of impulse transmission at the level of the neuro-muscular junction, with patients complaining of varying degrees of fatigue and muscle weakness. Palpebral ptosis is present in 90% of patients with MG [11].

A thymectomy is a surgical variant of drug therapy administered to patients with MG. In thymomatous forms of this condition, the tumor must be removed [12].

In the case of patients with nonthymomatous MG, thymectomy has become a therapeutic standard, despite the fact that there is no solid scientific evidence to explain its positive effect. There are two systematic reviews of the literature that recommend prospective studies follow this aspect [13,14].

Because of the potential for surgery to worsen the MG symptoms, it should only be performed when the patient is neurologically balanced [15].

Patients showing signs of increased muscle weakness, bulbar manifestations, or respiratory impairment should benefit from plasmapheresis in the preoperative period [15].

As a condition with a low incidence, MG is considered a rare, autoimmune disease with a complex pathophysiology [16,17]. From a clinical point of view, the condition can be divided into 4 classes: class I (ocular myasthenia), class II (generalized myasthenia), class III (fulminant acute myasthenia), and class IV (severe myasthenia late) [18,19].

The therapeutic approach of patients with MG consists of the administration of drug treatment, the purification of antibodies by plasmapheresis, or surgical intervention by thymectomy [19,20].

Improvement of symptoms may or may not occur. In the case of patients with associated morbidities, which present a high operative risk, radiotherapy can be considered as an alternative [21,22].

Minimally invasive surgery has resulted in a decreased length of hospital stay for patients with MG [23]. They also led to an important decrease in the number and intensity of postoperative complications [24,25].

The purpose of this paper is to synthesize the information presented in the literature in order to create an overview of the perspectives that the existing studies open in relation to the particularities of thymectomy. The criteria for the selected articles included the use of keywords (thymus, thymectomy, acetylcholine, myasthenia gravis, thymoma, thymomatous, and nonthymomatous) and publication data within the last 30 years. We have addressed the following databases: PubMed, Scopus, Web of Science, and Orphanet.

## 2. Indications for Thymectomy

One of the topics debated in recent years in relation to MG, without reaching a consensus yet, is related to the indications of thymectomy for MG [26].

Before discussing the types of surgical approaches, and even the superiority of thymectomy over medical treatment, one more issue must be considered [27].

Patient age, gender, presence of thymoma, specific seropositivity, and severity of MG symptoms are the aspects to consider for the management of thymectomy [26]. As a rule, both the operative indication, the preoperative evaluation, and the postoperative care should be managed with the cooperation of the surgeon, anesthesiologist, and neurologist [27].

In most cases of thymoma, patients should undergo thymectomy, regardless of whether MG is generalized, bulbar, or ocular. A complete resection of the thymoma should be aimed for. If this is not possible, medical treatment can be administered both to relieve myasthenic symptoms and to prevent local invasion. It has been reported that the remission and recovery rates of patients with MG associated with thymoma are similar or slightly worse than in nonthymomatous patients [28] (Table 1).

In the absence of thymoma, thymectomy may be indicated in cases of generalized MG with a positive specific serology [29,30].

The response to thymectomy does not appear to be influenced by the severity of MG [31].

The prevalence of remission of MG after thymectomy is not related to gender or age [32].

It must, however, be taken into account that elderly patients may have a weaker response to thymectomy due to their high degree of thymic involution. Considering this aspect, and the fact that operative risks are higher in elderly patients, most surgeons avoid thymectomy in this category of patients. However, some authors suggest the individual assessment of patients by evaluating the benefits and risks and consider that advanced age is not a factor in excluding this category of patients from the benefits of a thymectomy [33].

The role of thymectomy in young patients is not fully known. However, thymectomy may be considered in children with generalized MG with positive acetylcholine receptor anti-receptor serology who do not respond satisfactorily to pyridostigmine therapy or immunosuppressive therapy. There may also be an indication for thymectomy in children for whom it is necessary to prevent possible complications of immunosuppressive therapy [34].

The role of thymectomy in patients with double-seronegative MG is not fully known. However, most clinicians recommend thymectomy even in these cases. There are studies that highlight the fact that there are comparable results in terms of efficiency, both in patients with negative acetylcholine receptor antibody serology and in patients with positive specific serology [35].

In children with generalized seronegative MG, the feasibility of thymectomy should be evaluated [36].

MG may present an increased risk of exacerbation of symptoms during pregnancy. In particular situations, the occurrence of an acute myasthenic crisis syndrome in pregnant patients could occur. However, thymectomy should be avoided and delayed until after delivery due to the significant risks this surgery may pose to the mother or the fetus [37].

## 3. Limits and Risks of Surgical Intervention

Patients with MG are at significant risk of having a thymectomy because of the impairment of respiratory function during the disease [38].

However, with the advancement of operative techniques and improved performance in the fields of anesthesia and intensive care, the risks of a thymectomy are now manageable [39].

The mortality rate associated with thymectomy is less than 1%, even in patients with poorly controlled MG [39].

Complications related to thymotomy may occur due to the following conditions: acute myasthenic crisis, nosocomial infections, lesions of the recurrent laryngeal nerves, or phrenic nerves [40].

## 4. Types of Surgical Approach in Thymectomy

The goal of thymectomy is to remove as much of the thymic tissue as possible. Considering that the mediastinal and cervical adipose tissue may contain traces of thymic cells in varying amounts, the surgical approach should aim to achieve a resection as extensive as possible, avoiding damage to the recurrent laryngeal, left vagus, or phrenic nerves [41].

To achieve these objectives, there is the possibility of choosing between four major surgical techniques:(a)Transcervical thymectomy,(b)Minimally invasive thymectomy (video-assisted or robotic);(c)Trans-sternal thymectomy;(d)Combined trans-cervical–trans-sternal thymectomy [42].

In all of these procedures, the thymus is resected, but the resection of extracapsular mediastinal and cervical adipose tissue varies. There is no convincing evidence of superior efficacy or long-term remission rates in patients with MG for any of these surgical approaches [42].

Median sternotomy is preferred by most of the surgeons. This approach provides a wide area of exploration from the mediastinum to the neck, allowing complete resection of all thymic and associated adipose tissue [42].

Some authors support the effectiveness of an extended cervical thymectomy, taking into account less postoperative pain. This surgical approach leads to a period of hospitalization approximately equal to that required for patients who benefit from a median sternotomy, but the incisions made in this case are much smaller [43,44].

A special manubral retractor has been developed to improve exposure of the mediastinum and facilitate resection. The controversial element in relation to this approach is related to the impossibility of complete discovery, from an anatomical point of view, of the thymus, which leads to the risk of residual thymic tissue in the posterior part of the mediastinum. However, recent studies have reported comparable results to those obtained in the case of patients who underwent surgical interventions with a median sternotomy approach [45,46].

The simple transcervical approach is rarely preferred because of the risk of remnant thymic tissue in most patients [46].

Thymectomy by minimally invasive procedures such as video-assisted thoracoscopy or robot-assisted approaches is associated with low morbidity and mortality rates [47,48].

There is no doubt that minimally invasive approaches have lower morbidity and shorter hospital stays than more invasive approaches [49].

In addition to the obvious advantage given by the possibility of avoiding sternotomy, video-assisted approach techniques have similar results to those obtained by choosing median sternotomy. Also, a great advantage in the case of video-assisted mediastinal exploration techniques is represented by a much lower rate of postoperative complications, which may occur following thymectomy [50].

The rate of surgical approaches involving the use of video-thoracoscopy or robotic-assisted surgery is increasingly preferred for performing thymectomy in patients with MG. According to the studies carried out so far, the types of minimally invasive thoracic approaches for thymectomy offer comparable results to those obtained by using more aggressive techniques [51].

In order to highlight the results of the thymectomy literature review, in the cases of patients with myasthenic manifestations, 146 papers were analyzed, of which 16 were selected, and their results are presented in Table 2.

The main condition for the inclusion of the articles in our analysis was represented by the existence of a measurement of the results obtained in patients with myasthenia gravis following the use of at least one thymectomy technique. Also, “case report” and “review” type articles were excluded. Another inclusion criterion was the need for the subjects included in the study to be adults. Thus, papers that included results from pediatric patients were not discussed.

## 5. Determining the Moment to Perform the Thymectomy and the Preoperative Preparation of the Patient

Thymoma patients should be evaluated for surgical treatment without delay. However, the optimal time for performing thymectomy has not been determined in patients with MG who do not present thymoma [66].

The most suitable patient for thymectomy is one with minimal bulbar or respiratory symptoms. This must be taken into account to avoid perioperative complications as well as to reduce the doses of corticoids as the time of surgery approaches and to minimize the risk of postoperative infection. Some studies support the idea that thymectomy is more effective in the early stages of MG. This fact is due to the better remission rates of MG in patients with early stages of the disease [67].

Although early thymectomy has no proven benefit, it is mostly recommended to be performed within the first 3 years after the onset of the condition [68].

Any patient with MG for whom a thymectomy is considered should undergo a thoracic computed tomography with a contrast agent. This investigation is all the more useful in patients with thymoma for the evaluation of a mediastinal mass and possible associated vascular invasions. Obtaining a histopathological diagnosis by transthoracic needle aspiration is still controversial [69].

Patients with MG scheduled for thymectomy should be operated on when the clinical condition is optimal. In the preoperative evaluation stage of patients with MG, the following parameters related to the specifics of the disease must be taken into account:-Recent evolution of the disease,-Degree of muscle weakness,-Drug treatments included in the patient’s therapy at the time of evaluation,-Comorbidities,-Pulmonary function [67].

Interdisciplinary collaboration is extremely important in the approach of a patient with MG who is about to undergo thymectomy. A detailed evaluation of pulmonary function should be performed preoperatively, which could provide an appropriate prognosis related to postoperative care. In patients with bulbar or respiratory symptoms, in the preoperative stage, immunoglobulin therapy or its association with plasmapheresis is necessary. This will also help to reduce the amount of corticosteroids required for drug therapy, both preoperatively and postoperatively, in any patient with MG undergoing thymectomy [70].

Plasmapheresis significantly improves both respiratory function and muscle strength in patients with MG undergoing thymectomy. Also, this therapeutic technique significantly reduces the hospitalization period [71].

Continuation of anticholinesterase therapy until surgery is essential, usually by the morning of the surgery. Also, the administration of anticholinesterases is useful in the postoperative stage, right from the moment patients regain consciousness, because it is extremely important to avoid the appearance of oropharyngeal or respiratory symptoms [72].

## 6. Postoperative Follow-Up of Thymectomized Patients

Following the operation, the patient is awake and closely monitored by an intensive care specialist. Extubation is performed if respiratory function and blood tests are appropriate. In almost all cases, extubation can be performed early. The postoperative follow-up of the thymectomized patient must be carried out by a multidisciplinary team formed by a surgeon, neurologist, and intensive care specialist. If acute respiratory failure occurs, reintubation should be performed urgently [73].

To assess respiratory status, the vital capacity of the lungs should be assessed every 6 h by inspiratory-expiratory pressure measurements. Aggressive bronchopulmonary clearance measures should also be taken [74].

Early initiation of anticholinesterase therapy in the postoperative period reduces the risk of oral and tracheal secretion problems. By doing so, the risk of a cholinergic crisis may be minimal. If respiratory malfunctions occur in the postoperative period in thymectomized patients, the initiation of plasmapheresis therapy should be considered in absolute emergency conditions. The transfer of the patient from the intensive care unit should be carried out only in a situation where he is stable from a respiratory point of view. It is also important to favor the early removal of drain tubes, taking into account the comfort of the patients and decreasing the risk of infection [75].

There are situations in which a postoperative exacerbation of myasthenic symptoms may occur. There are a number of factors that influence the occurrence of postoperative myasthenic crisis, associated with the need to ensure the continuation of mechanical ventilation in thymectomized patients:-Weakness of respiratory muscles,-A vital pulmonary capacity of less than 2 L,-Bulbar manifestations,-History of the myasthenic crisis,-A serum level of anti-acetylcholine receptor antibodies exceeding 100 nmol/L,-Intraoperative blood loss greater than 1 L [76].

Patients presenting such prognostic factors require increased attention in the postoperative care stage [77].

Cholinergic crisis develop as a result of overstimulation of nicotinic and muscarinic receptors at the level of neuromuscular synapse junctions. This phenomenon usually occurs secondary to the inactivation or inhibition of acetylcholinesterase (AChE), the enzyme responsible for releasing acetylcholine [78].

The excessive accumulation of acetylcholine in the neuromuscular junctions causes the symptoms of muscarinic and nicotinic toxicity, such as muscle cramps, excessive salivation, lacrimation, muscle weakness, paralysis, diarrhea, and vertigo [79].

Cholinergic crisis occurs in the following situations:-In the case of patients with MG, who receive treatment with acetylcholinesterase inhibitors in high doses;-In the case of patients who are under post-general anesthesia and have received high doses of acetylcholinesterase inhibitors to neutralize neuromuscular blocking agents during the intervention, such as neostigmine;-In case of exposure to a chemical substance that causes the inactivation of acetylcholinesterase: sarin gas, some pesticides, and insecticides [80].

Myasthenic crisis is a complication of MG. Among the triggers of myasthenic crisis are: infections, surgical interventions, menstruation, and some drugs (quinidine, calcium channel blockers, and some antibiotics) [81].

The clinical symptomatology of myasthenic crisis is very similar to that of cholinergic crisis. Cholinergic crisis should be evaluated in every case of myasthenic crisis, although they are not completely similar [82].

It is important to identify the type of seizure that is causing the muscle weakness. This is achieved by administering an IV dose of 2 mg of edrophonium, which produces an improvement in the clinical symptoms in myasthenic crisis, whereas in patients who are in cholinergic crisis, the symptoms are aggravated by this drug [83].

The treatment of acute myasthenic crisis consists of the administration of rapid immunotherapy with immunoglobulins administered intravenously, or by using plasmapheresis. During this time, patients should be evaluated for possible infection or other complications, such as the use of medications that may exacerbate MG attacks [84].

In case of immunotherapy or plasmapheresis with limited results within a few weeks, prednisone or methylprednisolone must be administered. Although cholinesterase inhibitors are also available for IV administration, they should be avoided in a crisis because they can increase respiratory tree secretions, complicating airway management. Therefore, all acetylcholinesterase inhibitors will be withheld throughout mechanical ventilation [85].

Regardless of the stage of MG, all cases associated with thymoma should be operated on by resection. If complete excision of the thymoma is not possible, radiotherapy and chemotherapy should be performed both to control myasthenic symptoms and to prevent local invasion. Thymectomy is recommended for patients younger than 60 years with generalized, non-thymomatous MG and anti-acetylcholine receptor antibodies. Plasmapheresis or intravenous immunoglobulin is recommended before thymectomy in patients with preoperative respiratory or bulbar symptoms [86,87].

## 7. VATS Thymectomy

Thymectomy by video-thoracoscopy (VATS) is necessarily performed under general anesthesia. To achieve this, first of all, selective endotracheal intubation is necessary [88].

Bilateral video-thoracoscopic thymectomy, especially the uniportal one, is a purely endoscopic procedure with direct access to the monitor. The surgical dissection is started to the left, with the patient seated in a lateral recumbent position at an angle of 60°. An incision of 3–5 cm is performed in the fourth intercostal space on the anterior axillary line without affecting the ribs. Thus, the lung is deflated, and through a minithoracotomy, the thoracoscope and conventional thoracoscopic instruments are inserted. The camera is placed at the rear end of the uniportal access. Dissection usually starts from the left pericardiophrenic angle. Proceeding cranially, the next step is to incise the mediastinal pleura along the anterior line of the phrenic nerve and mobilize the thymus and perithymic fat. Endoscopic vascular clips are used to control the thymic vessels. The dissection is performed in the pre-pericardial plane. Finally, the superior thymic poles are dissected gently and pulled down to achieve their complete mobilization until the thyroid-thymic ligaments are visible for sectioning. After finishing the left thymic dissection, we open the contralateral pleura to push the mobilized part of the thymus into the right pleural cavity while the lung is in apnea for a few seconds. At this point, the residual mediastinal fat is completely dissected from the left phrenic nerve, the innominate vein, and the left pericardiophrenic angle. A single tube is inserted through the incision and curved downward to the costophrenic angle [89,90].

The patient is then placed on the contralateral side, also in lateral recumbent, at an angle of 60°. Similarly, on the left side, a 3–5 cm incision is performed in the fourth intercostal space, at the level of the anterior axillary line, without damaging the ribs. The dissection starts from the right pericardiophrenic angle and continues cranially, along the anterior border of the phrenic nerve. The thymus is then mobilized from the surrounding tissue and from the superior vena cava. Vascular clips are used to secure the thymic veins draining into the superior vena cava. Finally, “en bloc” thymic dissection, with bilateral perithymic and pericardiophrenic fat tissue, is completed, and removed using a 15 cm endosac via uniportal access. At this operative step, the mediastinal fat is completely dissected from the right phrenic nerve, innominate vein, and right pericardiophrenic angle. A single chest drain tube is inserted through the incision, curved downward to the costophrenic angle. The lung is then re-inflated [91].

During the intervention, the surgeon and the assistant are positioned behind the patient. The camera is placed at the posterior end of the uniportal access and is controlled by the assistant. The camera does not interfere with the surgeon’s instruments and provides a stable view of the operative field. It is also mandatory that the surgeon’s action be in the center of the screen. Finally, synchronization between the surgeon and his assistant is essential to reducing the operative time and the success of the procedure [90]. This means there should be trained operative teams for this procedure.

Standardization of technique and maintaining the same surgical team could be very useful ways to achieve performance [92].

Bilateral uniportal video-assisted thoracoscopic extended thymectomy is a safe and feasible technique for surgical resection of thymic hyperplasia and thymoma. This technique could be considered a development of multiportal thoracoscopic thymectomy that offers all the advantages of minimally invasive surgery, such as:-Reduced postoperative pain,-Faster mobilization of patients,-Reduced hospitalization period,-Better results in terms of the postoperative scar [90].

## 8. Robotic Thymectomy

Robotic thymectomy can be performed on patients with normal thymuses or with benign and malignant thymus tumors. Invasion of the great vessels of the heart is generally a contraindication to minimally invasive thymectomy. Even large thymic tumors can be safely approached with robotic assistance [93].

The preoperative evaluation of the patient to undergo thymectomy depends on the course of MG. In this sense, it is essential to carry out a questionnaire aimed at the detailed history of the patient. It is also necessary to physically assess the patient, highlighting the presence of eyelid ptosis, diplopia, dysphagia, or fatigue. In patients with MG, an appropriate serological and electrophysiological evaluation and treatment should be performed before surgery to avoid perioperative complications. Patients in acute myasthenic crisis should not undergo emergency thymectomy. In these cases, the surgical intervention must be delayed, prioritizing overcoming the acute myasthenic crisis through anticholinesterase inhibitor drug therapy, immunotherapy, or plasmapheresis. Afterwards, an appointment can be made for a thymectomy [94].

The typical approach used in the case of robotic thymectomy is through minithoracotomy at the level of the right hemithorax. Thus, selective intubation is performed, and then the patient is positioned in the lateral recumbent position. After making the incision and positioning the port, the robotic arm is inserted at the level of the chest. The camera port is inserted midway between the sternal angle and the xiphoid process. During a robotic surgical procedure, the following aspects must be followed:-The complete resection of the surgical parts targeted by the surgical intervention is essential;-The most serious complication of the procedure is represented by the injury to the great blood vessels during the dissection performed in order to extirpate the thymus or thymomas;-The most common technical problem is the interaction of the robotic arm with the patient’s right shoulder. To avoid this problem, the shoulder should be positioned as low as possible. Placing the port for the robotic arm as anteriorly as possible and using a trocar for the long robotic arm are also helpful;-It is not mandatory to place the camera on the left side of the patient to visualize the left phrenic nerve. A 30° down chamber can be exchanged for a 0° chamber and used if the rib is not visible. If the nerve is masked by a tumor mass, an additional camera can be added, depending on the particularity of the case, on a new port, positioned on the left side [95].

According to some authors, anterior robotic dissection must follow four rules: “en bloc”; “no touch”; “stay and play”; “the right camera angle for each dissection area”. As a rule, a total resection of the thymus is used together with the thymus and the thymic fat. When the camera is placed correctly, the “stay and play” technique intervenes, i.e., everything that can be easily dissected is dissected. The 30° downward camera is used during dissection of the thymus from the pericardium, near the phrenic nerve, or in the contralateral pleural space cavity. The 30° upward chamber is used during dissection of the thymus from the sternum, from the opposite mediastinal pleura, at the angle between the left phrenic nerve and the sternum, and during dissection of the superior thymic poles. The “no touch” rule refers to avoiding tumor seeding [96].

By using robotic thymectomy, excellent perioperative and long-term results can be achieved. Most of the time, the number and degree of complications are reduced, with a hospital stay of 1–2 days and socio-professional reintegration within about 2 weeks.

Robotic thymectomy results in decreased blood loss, a shorter time for drain tubes to be maintained, and last but not least, a shorter hospital stay [97].

Also, the level of postoperative pain felt, the return to functional parameters, and the quality of life are clearly superior compared to the results obtained following the application of surgical techniques using midline sternotomy.

Most of the time, the biggest disadvantage of robotic thymectomy is represented by the increased cost that this technique requires [98].

The neurological results obtained following the use of robotic thymectomy, followed 5 years after the surgical intervention, led to complete remission in approximately 29% of cases and improvement of the clinical symptoms in 87.9% of the cases studied [99].

In the case of patients with MG and associated thymoma, the long-term results of the studies show a 5-year survival rate of 90%, although there is no clear data regarding the recurrence rate in the case of these patients [100].

Even in pediatric patients, as well as in patients with large tumors involving the pericardium, lung, or phrenic nerve, robotic thymectomy has been performed safely with very good results [100].

The only indication for the conversion of the surgical intervention to a classical technique is represented by the invasion of the mediastinal vessels. Thus, it can be considered that there are important advantages to the use of robotic thymectomy techniques, among which we mention:-The ability to have a three-dimensionally thoracic view, during the surgical procedure;-Increasing the dexterity and precision of the specific maneuvers;-An easier and better approach to the anterior mediastinum [100].

Because robotic thymectomy can be performed safely and with minimal perioperative morbidity, it has become more common worldwide [101].

## 9. Discussions

Thymectomy indications are a sensitive subject, for which reaching a consensus is difficult. Thymectomy can be indicated both in the presence and in the absence of thymoma.

In myasthenic syndromes determined by the existence of a thymoma, only age seems to influence the response to thymectomy, while other characteristics, such as sex, do not seem to have any influence. For this reason, the indications for thymectomy in a geriatric patient are different.

So far, four main methods of thymectomy have been described: transcevical, transsternal, combined, and minimally invasive. Regardless of the technique used, the goal is total resection of the thymus. The only aspect that varies is related to the resection of thymic fat. With the passage of time, an important increase in the use of minimally invasive methods of thymectomy has been observed. This is notable, especially given that thymectomy by minimally invasive methods in myasthenic patients seems to have similar results to thymectomy performed by classical surgery.

The discovery of VATS techniques brought new outcomes to thoracic surgery. This technique, which uses a thoracoscope inserted through a small incision in the chest wall, maximizes muscle and tissue preservation. Due to the low morbidity and mortality rate, this is currently the main technique in most thoracotomies. Compared to classical surgery, the results of VATS are similar or even superior to those of open surgery [90].

The technique of video-assisted thoracoscopy is represented by the dissection of veins, arteries, and pulmonary bronchi. These often combine mediastinal lymphadenectomy using a one- to four-port approach. In this way, an image visualized on a screen is obtained without the need to move the ribs apart [101].

Although additional training of the surgeon is required, VATS can provide a different exposure of the intra-thoracic organs and provide an improved visual perspective. Although the tendency is to use as few ports as possible in video-assisted surgery, it is very important that the main goal is efficient resection and not minimization of chest wounds. Also, whenever necessary, it is important that the surgical team is ready to transform the surgical intervention into a classic approach, with the aim of performing the resection in a correct manner.

Even under these conditions, there is a continuous evolution in what it means to minimize the wound at the level of the chest wall, and uniportal VATS (spVATS) has become more and more common for lung resection.

The subxiphoid approach via VATS was introduced as an alternative to VATS with an intercostal approach because avoiding the intercostal incision leads to reduced pain and facilitates early mobilization and improved recovery. Access to the pleural cavities and the anterior mediastinal space through a subxiphoid incision presents some challenges, but is useful in thymectomy [102].

It is also important to observe the evolution of the results related to the clinical condition of patients with myasthenia gravis who benefit from thymectomy. The information presented in Table 1 highlights a quasi-constant improvement in these results.

Considering all these aspects, it can be stated that thymectomy includes a complex group of surgical techniques that are constantly evolving and have the potential to improve the results obtained by administering therapies to patients with thymic damage.

Also, the evolution of robotic surgery, which can greatly improve the prognosis and quality of life of these patients, must be evident.

To determine the operative moment, it is necessary to evaluate several aspects related to the patient’s condition. From this point of view, it is important to take into account a series of factors: the degree of muscle weakness, the associated comorbidities, the pulmonary function, the drug treatments administered preoperatively, as well as the recent evolution of the disease. From this point of view, in patients in whom it was necessary to improve respiratory function, plasmapheresis proved extremely useful. Besides this, in most cases, the continuation of anticholinesterase therapy is essential.

From the point of view of postoperative evolution, the most serious situations are those represented by the occurrence of acute myasthenic crisis. An increase in vigilance is necessary in the case of patients who present a high risk for such an event. In the event of this situation, plasmapheresis and the administration of immunotherapy are essential.

The newest method of surgical approach to the thymus is represented by robotic surgery. This presents important advantages regarding the patient’s recovery and, of course, increases the efficiency of the operation in many cases. From the point of view of the operative indications, robotic surgery can be performed in almost all situations where another type of thymectomy is indicated. From this point of view, this thymectomy technique is becoming more and more preferred. However, the main disadvantage is represented by its high costs.

## 10. Conclusions

This paper presents an up-to-date literature synthesis about thymectomy. Special attention has been paid to VATS and robotic thymectomy, as they are increasingly used. Although this study presents an overview of the types of thymectomy, not enough information was found regarding how thymectomy succeeds in improving the clinical status of some categories of patients. Such an example refers to patients with MG and negative serology. This work opens new perspectives in relation to the study of thymectomy as a treatment for patients with seronegative myasthenia gravis. Other perspectives that this work opens are related to the role that the use of new robotic or video-assisted thymectomy techniques could have in relation to the dynamics of AchR antibody concentrations by reducing the time spent in the intensive care unit.

## Figures and Tables

**Table 1 jpm-14-00241-t001:** Thymectomy indications.

Pacients	Indications	Comments
Thymomatous	- Most of the patients—taking into consideration the staging	- The complete resection of the thymoma must be followed - MG remission is comparable to that of nonthymomatous patients
Nonthymomatous	- Patients presenting MG with positive specific serology - Young patients with generalized MG, who do not respond to drug treatment - In pediatric patients in whom it is necessary to avoid immunosuppressive treatment	- Elderly patients may have a weaker response to thymectomy - The role of thymectomy in young patients is not fully known

**Table 2 jpm-14-00241-t002:** Relevant research studies in the evaluation of different approach techniques in thymectomized patients.

Authors	Number of Subjects	Group Characteristics	Types of Approach	Results of the Intervention	Type of Study
Frist et al. (1994) [52]	46	Patients with and nonthymomatous included in the study	Thymectomy through sternotomy	28% complete remission	Retrospective study
Masaoka et al. (1996) [53]	286	Patients with nonthymomatous MG	Thymectomy through sternotomy	55.7% complete remission 10 years postoperatively	Retrospective study
Venuta et al. (1999) [54]	217	Patients with nonthymomatous MG	Transcervical thymectomy and sternotomy	46% complete remission at 18 months postoperatively	Retrospective study
Budde et al. (2001) [55]	113	Patients with thymoma and nonthymomatous included in the study	Thymectomy through sternotomy in 84% of cases	Different degrees of remission in 75% of patients in the postoperative period	Retrospective study
De Perrot et al. (2001) [56]	35	Patients with nonthymomatous MG	Transcervical approach	Different degrees of remission in 88% of patients in the postoperative period	Retrospective study
Mantegazza et al. (2003) [57]	206	Patients with nonthymomatous MG	VATS	Complete remission in 53.9% of patients at 6 years	Prospective study
Tansel et al. (2003) [58]	204	Patients with nonthymomatous MG	Thymectomy through median sternotomy	The remission rate in different degrees was 72% at 1 year	Retrospective study
El-Medany et al. (2003) [59]	100	7 patients with thymoma and 93 nonthymomatos included in the study	Maximal thymectomy (combined approach—transcervical and transsternal	Complete remission was observed in 75% of subjects 15 years postoperatively	Retrospective study
Kawaguchi et al. (2007) [60]	34	Patients with nonthymomatous MG	20 patients benefited from thymectomy (different types and approaches) 14 patients benefited only from drug therapy	Clinical remission was observed in 50% of patients with thymectomy vs. 17% in patients treated only with medication	Retrospective study
Pompeo et al. (2009) [61]	32	Patients with nonthymomatous MG	VATS	VATS thymectomy has highly satisfactory long-term results in non-thymomatous MG, with a 10-year remission rate of 50% and an overall response rate of 90%	Retrospective study
Lin et al. (2010) [62]	60	Patients with nonthymomatous MG	Transsternal thymectomy and VATS	The complete remission rate was not influenced by the type of approach, being approximately 32% at 38.5 months in both cases	Retrospective study
Spillane et al. (2013) [63]	89	Patients with thymoma and nonthymomatous included in the study	Extended transsternal thymectomy	The need for corticosteroid administration decreased from 73% of cases preoperatively to 47% postoperatively	Retrospective study
Voulaz et al. (2018) [64]	157	Patients with thymoma included in the study	VATS thymectomy was attempted in 34 cases. 123 open surgery tehiques.	Five and ten-year disease-free survival rates were 91.1% (radical thymectomy) and 81.8% (conservative)	Retrospective study
Cabrera-Masqueda et al. (2020) [65]	46	Patients with thymoma and nonthymomatous included in the study	84.8% VATS 15.2% open surgery.	After ten years of follow-up, 9.8% reached complete stable remission, a total of 32 patients (78%) had a favourable outcome and thymoma was not correlated	Retrospective study

## Data Availability

Data are contained within the article.

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
