# Peer review of "Thymus Surgery Prospectives and Perspectives in Myasthenia Gravis"

_jpm, 2024, doi:10.3390/jpm14030241_

Round 1

Reviewer 1 Report

Comments and Suggestions for Authors

The title is interesting and captivates reader's attention. Introduction provides sufficient information, considering the length of this article and the detailed data offered in the body of the text. The article is relatively well structured, covering and synthesising all thymus surgery for Myasthenia Gravis aspects. Though, the literature review provided by the authors should be constructed more academically, stating exactly the purpose of the search in the 123 papers analysed and the specific criteria used for selecting only 14 papers that were presented in the manuscript. Moreover, the review of the literature should be integrated in the body of the text and not in the discussion. References are mixed, both up to date and old ones, which is not necessarily wrong, but I considere that all this valuable information should be presented more organised, so that the reader can clearly understand the actual trends in the surgical management of Myasthenia graves separately by what has been done historically. In this regard, the discussion can be improved.

Comments on the Quality of English Language

Moderate English editing is required, as well as orthographic and spelling revisions.

Author Response

Thank you very much for taking the time to review this manuscript. Please find the detailed responses below.

  1. Though, the literature review provided by the authors should be constructed more academically, stating exactly the purpose of the search in the 123 papers analysed and the specific criteria used for selecting only 14 papers that were presented in the manuscript.

Thank you for pointing this out. We agree with this comment, thus we introduced in the text the criteria that were the basis of the selection of the 14 works presented in the manuscript.

  1. Moreover, the review of the literature should be integrated in the body of the text and not in the discussion.

We agree with this aspect, so we have repositioned this part of the manuscript in the "Types of surgical approach in thymectomy" section.

  1. References are mixed, both up to date and old ones, which is not necessarily wrong, but I considere that all this valuable information should be presented more organised, so that the reader can clearly understand the actual trends in the surgical management of Myasthenia graves separately by what has been done historically. In this regard, the discussion can be improved.

We agree with the fact that it is important for the reader to be able to understand more clearly the evolution in the management of myasthenia gravis. For this reason, we reorganized the table and tried to redo the discussion chapter, emphasizing the presentation of the evolution of the thymectomy study.

Reviewer 2 Report

Comments and Suggestions for Authors

Thank you for the opportunity to analyze your review.       

In this article, authors reviewed the place of thymectomy in Myasthenia Gravis. 

            Concerning the introduction:

            The introduction is well written, with a good overview of the physiopathology of myasthenia gravis (MG) and its 2 forms: Thymomatous and nonthymomatous. Maybe these 2 forms can be described as more different. 

No major concerns except:

Line 76: Videothoracoscopic surgery or robotic surgery -> Minimally invasive surgery

            Concerning the Methodology:

            Why not having included the 2 word thymomatous and nonthymomatous also? 

Concerning the 2nd paragraph: “Indications for thymectomy”

            No major concerns except:

            Lines 92 and 93: “the preoperative evaluation and the postoperative care should be managed by the cooperation of the surgeon and the neurologist (27).” And the anesthesiologist?  

            Lines 116 and 117: Immunosuppressive’s complications prevention and corticoid sparing are major indications to perform a thymectomy. For Children, but also for young patients and elderly also. 

            Maybe thymectomy indications can be synthetized in a table. 

            Concerning the 3rd paragraph: “Limits and risks of surgical intervention”

            You must discuss complications related to MG, and complications related to minimally invasive surgery. This paragraph needs to be more detailed. 

            Maybe this can be placed after the 4th paragraph. 

            Line 138: thymotomy should be corrected.

            Concerning the 4th paragraph: “Types of surgical approach in thymectomy”

As you described before, MG can be related to a thymoma or not. So, for MG thymomatous MG some authors and surgeons discussed a thymomectomy associated to thymectomy with a less extensive dissection compared to a complete thymectomy for nonthymomatous MG. This need to be presented. 

Lines 157 to 159, ref 42: In 2024, Minimally invasive approach became the gold standard approach to perform this surgery, but I agree with you there is a lack of consensus and evidence-based data. This is the conclusion of this paragraph lines 185 to 187, and you should start this paragraph by minimally invasive approaches and the open approaches. 

Finally in the 7th paragraph you are dealing with VATS Thymectomy. Why not in the same paragraph?

But dealing with surgery, moreover, the type of surgical approach you need to discuss “the minimally invasive approaches” that are described as sequential and bilateral, subxiphoid…

Concerning the paragraph 5: “Determining the moment to perform the thymectomy and the preoperative preparation of the patient”

Thymoma is a cancer. So, priority is to stabilize MG, in order to early remove the thymoma.

For nonthymomatous MG, better results are observed for patients who are early treated and surgically managed. 

For pre operative management to stabilize the MG, unfortunately few consensuses are reported and its mainly a regional protocol. 

Concerning the 6th paragraph: “Postoperative follow-up of thymectomized patients”

Maybe a too long paragraph for finally few informations. 

Lines 273 and 274: Is it a condition often found in clinical practice? 

Concerning the paragraph 7: “VATS thymectomy”

Dealing with the anesthetic monitoring, urinary catheter and arterial catheter are not used in routine in many surgical teams. You have based your description from an old papier published in 2002!

In this paragraph, your references are not from surgical reports for many of them. Only 2 on 6. One of your references is about subxiphoid approach but you have described a bilateral uniportal approach.

Confusing paragraph.

References are not up to date.

Concerning the paragraph 8: “Robotic thymectomy”

Lines 358 to 367: It’s not only for a robotic approach. It’s general conditions for all the surgical approach for a thymectomy. 

Most of the surgical teams are perfoming robotic thymectomy from a left-sided approach in order to identify the phrenic nerve. 

You are bringing some tips and tricks for a robotic thymectomy, but some references are missing. 

Line 395, need to insert the correct reference. 

References are not up to date.

Dealing with surgery, with surgical technical details, tips and tricks, you can use references from reports dealing with thymectomy for non MG patients. 

Results of thymectomy are just for robotic approach? Not VATS or open? 

Concerning the paragraph 9: “Discussion”

I’m a little bit astonished about your article selection. Many old references. 

What did you discuss in the discussion?

Concerning the conclusion:

            Finally, it’s a long article, with many, many references but not sure that they are all relevant? Moreover, many reports were published a long time ago. References are not up to date? 

            Moreover, in a recent paper published in the NEJM, authors discussed long term consequences of thymectomy.  Need to deals with this new topic of interest. 

            This review can be more detailed and also shorten to bring up to date informations. 

Typo:

Line 64: There are two systematic reviews of the s literature

Comments on the Quality of English Language

Minor editing of English language required

Author Response

Thank you very much for taking the time to review this manuscript. Please find the detailed responses below.

  1. Concerning the introduction:

            The introduction is well written, with a good overview of the physiopathology of myasthenia gravis (MG) and its 2 forms: Thymomatous and nonthymomatous. Maybe these 2 forms can be described as more different. 

No major concerns except:

Line 76: Videothoracoscopic surgery or robotic surgery -> Minimally invasive surgery

Thank you for pointing this out. We believe that this change contributes to the creation of a more concise text.

  1. Concerning the Methodology:

            Why not having included the 2 word thymomatous and nonthymomatous also? 

Indeed, the inclusion of the two keywords leads to the improvement of the results obtained in the search for relevant works for this review, which are found in the current form of this manuscript.

  1. Concerning the 2ndparagraph: “Indications for thymectomy”

            No major concerns except:

            Lines 92 and 93: “the preoperative evaluation and the postoperative care should be managed by the cooperation of the surgeon and the neurologist (27).” And the anesthesiologist?   Lines 116 and 117: Immunosuppressive’s complications prevention and corticoid sparing are major indications to perform a thymectomy. For Children, but also for young patients and elderly also. Maybe thymectomy indications can be synthetized in a table. 

We agree that the suggested changes improve the organization of the information in this paper, so they can be found in the current text, together with table 1, which summarizes the indication of thymectomy.

  1. Concerning the paragraph 7: “VATS thymectomy”

Dealing with the anesthetic monitoring, urinary catheter and arterial catheter are not used in routine in many surgical teams. You have based your description from an old papier published in 2002! In this paragraph, your references are not from surgical reports for many of them. Only 2 on 6. One of your references is about subxiphoid approach but you have described a bilateral uniportal approach. Confusing paragraph. References are not up to date.

Thank you for pointing this out. Indeed, the bibliographic references in this paragraph were entered incorrectly. I redid the bibliography for this part of the manuscript.

  1. Concerning the paragraph 8: “Robotic thymectomy” Lines 358 to 367: It’s not only for a robotic approach. It’s general conditions for all the surgical approach for a thymectomy. Most of the surgical teams are perfoming robotic thymectomy from a left-sided approach in order to identify the phrenic nerve. You are bringing some tips and tricks for a robotic thymectomy, but some references are missing. Line 395, need to insert the correct reference. References are not up to date. Dealing with surgery, with surgical technical details, tips and tricks, you can use references from reports dealing with thymectomy for non MG patients. Results of thymectomy are just for robotic approach? Not VATS or open? 

Thank you for rating this paragraph. We updated the bibliographic reference in the case of line 395. Regarding the presented results, we emphasized the results obtained by robotic surgery, because we wanted to highlight the evolution of thymectomy with the appearance and development of this technique. We also considered it appropriate to point out these advantages of robotic surgery because it is the newest and most unexplored technique in terms of the results obtained.

  1. Concerning the paragraph 9: “Discussion” I’m a little bit astonished about your article selection. Many old references. What did you discuss in the discussion?

We agree that paragraph 9: "Discussions" should be revised, so we tried to present things differently, with an emphasis on the evolution of thymectomy techniques over time.

  1. Concerning the conclusion:     Finally, it’s a long article, with many, many references but not sure that they are all relevant? Moreover, many reports were published a long time ago. References are not up to date? Moreover, in a recent paper published in the NEJM, authors discussed long term consequences of thymectomy.  Need to deals with this new topic of interest.  This review can be more detailed and also shorten to bring up to date informations. 

             We agree that the bibliographic references include an important number of old titles, for this reason we tried to revise as much as possible the bibliography of this manuscript. However, considering that we tried to discuss including thymectomy techniques that are less used today, some of the references are still useful.

  1. Typo: Line 64: There are two systematic reviews of the s literature

Thank you for noticing this typo. This has been corrected.

Round 2

Reviewer 2 Report

Comments and Suggestions for Authors

Concerning the abstract :

« This manuscript opens the door in research possible physio-pathological correlations that the parathyroid gland could have an impact over both thymus pathology and myasthenia gravis. » -> This sentence seems to “arrive” without any question asked earlier about parathyroid gland! Need to correct that in the abstract. 

Moreover, nothing is described about parathyroid gland in all your article. If you deal with thymic gland, keep it simple and name it like that, not with another word. 

  1. Concerning the introduction:

No major concerns 

  1. Concerning the Methodology:

No major concerns 

  1. Concerning the 2ndparagraph: “Indications for thymectomy”

Table 1 is interesting. But you can write nonthymomatous rather than “without thymoma”. 

  1. Concerning the paragraph 7: “VATS thymectomy”

Thank you for your correction of the bibliography, but concerning the urinary catheter and the arterial monitoring it’s not used in routine even if for MG patients. 

5.     Concerning the paragraph 8: “Robotic thymectomy”

I noticed your comments.

But you can cite a recent article dealing with Robotic thymectomy for MG patients are highlighted 4 important surgical rules: en bloc”; “no touch”; “stay and play”; “the right camera angle for each dissection area”; J Vis Surg 2021;7:20 | http://dx.doi.org/10.21037/jovs-20-128;

No other major concerns

  1. Concerning the paragraph 9: “Discussion” I’m a little bit astonished about your article selection. Many old references. What did you discuss in the discussion?

Your corrections are welcome. The discussion is more uptodate to daily practice and questions reased by the surgical management of MG patients. But it can be synthtized a little bit more. 

  1. Concerning the conclusion: Finally, it’s a long article, with many, many references but not sure that they are all relevant? Moreover, many reports were published a long time ago. References are not up to date? Moreover, in a recent paper published in the NEJM, authors discussed long term consequences of thymectomy. Need to deals with this new topic of interest. This review can be more detailed and also shorten to bring up to date informations. 

We agree that the bibliographic references include an important number of old titles, for this reason we tried to revise as much as possible the bibliography of this manuscript. However, considering that we tried to discuss including thymectomy techniques that are less used today, some of the references are still useful.

Author Response

Thank you very much for taking the time to review this manuscript.
  1. Concerning the 2ndparagraph: “Indications for thymectomy”

Table 1 is interesting. But you can write nonthymomatous rather than “without thymoma”. 

Thank you for your comment. I changed all the situations where "without thymoma" appears in the text, with nonthymomatous.

2. Concerning the paragraph 7: “VATS thymectomy”

Thank you for your correction of the bibliography, but concerning the urinary catheter and the arterial monitoring it’s not used in routine even if for MG patients. 

After you brought this to our attention, we brought the text to a form that more accurately describes the technique used. 

3. Concerning the paragraph 8: “Robotic thymectomy”

I noticed your comments. But you can cite a recent article dealing with Robotic thymectomy for MG patients are highlighted 4 important surgical rules: en bloc”; “no touch”; “stay and play”; “the right camera angle for each dissection area”; J Vis Surg 2021;7:20 | http://dx.doi.org/10.21037/jovs-20-128;

No other major concerns

Thank you for this comment. We have used the information indicated in your comment regarding  paragraph 8

4. Concerning the paragraph 9: “Discussion” I’m a little bit astonished about your article selection. Many old references. What did you discuss in the discussion?

Your corrections are welcome. The discussion is more uptodate to daily practice and questions reased by the surgical management of MG patients. But it can be synthtized a little bit more. 

Thank you for these comments, which certainly help us to significantly improve the quality of this manuscript. Also, regarding the cited references, We have done our best to revise and update them accordingly.